# Modelling and Optimisation of FDM-Printed Short Carbon Fibre-Reinforced Nylon Using CCF and RSM

**DOI:** 10.3390/polym17131872

**Published:** 2025-07-04

**Authors:** Qibin Fang, Jing Yu, Bowen Shi

**Affiliations:** 1Department of Mechanical Engineering, Shantou University, Shantou 515063, China; 24qbfang@stu.edu.cn; 2College of Mechanical Engineering and Automation, Liaoning University of Technology, Jinzhou 121001, China; shibowen850@163.com

**Keywords:** short carbon fibre reinforced nylon, RSM, FDM, multi-objective optimisation, entropy weighting method

## Abstract

Nylon reinforced with short carbon fibres exhibits superior mechanical properties. Its use as a feedstock for fused deposition modelling (FDM) can extend its applications to consumer goods and industrial products. To investigate the flexural and impact properties of the FDM-printed short carbon fibre-reinforced nylon, a central composite face-centred (CCF) design with four factors and three levels and the response surface method (RSM) were employed. The four primary process parameters are the extrusion and bed temperatures, printing speed, and layer thickness. The three investigated responses were the flexural strength, flexural modulus, and impact strength. Perturbation curves and contour plots were used to analyse the influences of the individual and two-way interactions of the response parameters, respectively. Second-order statistical models were constructed to predict and optimise the mechanical properties. The optimal comprehensive mechanical properties were determined using a desirability function combined with the entropy weighting method. The predicted results of best comprehensive mechanical properties are 169.881 MPa for the flexural strength, 9249.11 MPa for the flexural modulus, and 29.659 kJ∙m^−2^ for the impact strength, achieved under the parameter combination of extrusion temperature of 318 °C, bed temperature of 90 °C, printing speed of 30 mm∙s^−1^, and layer thickness of 0.1 mm. A small deviation between the predicted and experimental results indicated the high reliability of the proposed method. The optimal outcomes under the studied parameters showed higher robustness and integrity than previously reported results.

## 1. Introduction

Fused deposition modelling (FDM) or fused filament fabrication (FFF) has gained popularity among both amateurs and professionals because of its affordable equipment, various feedstocks, and user-friendly operation. They have been widely used in prototyping, medicine, and industrial manufacturing. However, limited by the layer-by-layer deposition principle and the characteristics of thermoplastics, FDM-printed parts exhibit high process sensitivity, strong anisotropy, and weak inter-bead and interlayer adhesion, which hinder their further application in engineering areas [1]. Generally, the tensile performance of FDM-printed thermoplastic parts is inferior to that of their injection-moulded counterparts, whereas the compressive performance of FDM-printed parts is marginally lower than that of their injection-moulded counterparts [2]. Notably, by optimising the printing parameters, the impact strength of FDM-printed parts was found to outperform that of their injection-moulded counterparts [3].

The presence of numerous parameters in FDM, such as the extrusion temperature, printing speed, bed temperature, layer height, building direction, and infill pattern, affect the properties of the final parts [4]. Although filament producers generally provide wide ranges of the recommended printing parameters it is difficult to achieve the expected performance of printed parts.

Several types of filament materials are used in FDM. Polylactic acid, polypropylene, and polyethylene are easily processable and inexpensive, but the thermal and mechanical properties of the fabricated parts are poor. Nylon/polyamide (PA), polyethylene terephthalate glycol-modified, and acrylonitrile butadiene styrene exhibit moderate thermal and mechanical properties and are used in various engineering applications [5]. For high-performance thermoplastics, such as polyether ether ketone, polyetherimide, and polysulfone, the printer should be equipped with a high-temperature extruder, printing bed, and heating chamber. Additionally, the high material cost of the high-performance filaments and low printing efficiency restrict their wide application in the industrial and medical fields. Among the various thermoplastics, nylon and its copolymers possess better chemical resistance, wear resistance, and higher mechanical strength owing to stronger interlayer adhesion at elevated temperatures [6]. Therefore, FDM-printed nylon is widely used in home appliances, white goods, and various industrial applications [7]. It is evident that incorporating reinforcements/fillers, such as Kevlar fibres, glass, carbon, and natural fibres/particles, into a pure polymer can significantly affect the mechanical properties of printed parts [7,8]. Nylon composites reinforced with carbon fibre possess high strength, rigidity, and low weight, which make them promising in the aerospace and automotive industries [9]. Although continuous carbon fibre-reinforced nylon printed via FDM can provide superior mechanical properties, close to those of its injection-moulded counterpart, printing continuous fibre-reinforced composites requires expensive printers that cannot produce accurate and sophisticated geometries [7,10]. Therefore, short fibre-reinforced composites are most widely used for FDM.

As in the printing pure thermoplastic filaments, the printing temperature is a critical parameter for printed thermoplastic composite filaments because it has a significant influence on the bonding strength of the reinforcement/polymeric matrix and the interlayer adhesion [11]. The nozzle temperature determines the molten state and fluidity of the filament, whereas the printing speed governs the cooling process. Consequently, both parameters are directly related to the heating history during the entire modelling process. The layer thickness also governs layer adhesion and surface quality (staircase effect), thus determining the strength and stiffness of the printed parts [12]. The platform temperature determines the initial layers that stick to the platform, thus significantly influencing the warpage and shrinkage of the entire printed part [13].

Many studies have investigated the tensile properties of short carbon fibre-reinforced nylon [14]. Regardless of the optimal parameter combinations and load direction, the tensile properties of FDM-printed parts are significantly lower than those of their injection-moulded counterparts. It is noteworthy that various industrial applications focus more on the bending and impact properties; for instance, aircraft interior components include seat brackets, overhead luggage racks, engine cowlings, automotive intake manifolds, vehicle door handles, rear-view mirror supports, electronic product casings, bicycle frames, and racquets. Toro et al. [2] designed a full factorial experiment with three parameters (layer thickness, infill pattern, and density) to analyse the mechanical properties of nylon-20% (wt.) carbon fibre. Their results showed that the optimal parameter combination of nozzle diameter of 0.8 mm, layer height of 0.2 mm, the concentric pattern, and 100% infill density achieved the best comprehensive mechanical properties. Beylergil et al. [3] employed the Taguchi method to predict the Charpy impact strength of polyamide-20% (wt.) carbon fibre composites during the FDM process. Their study involved four factors, namely, the nozzle temperature, printing speed, infill density, and raster angle, and yielded an increase of 149.5% in impact strength compared to the highest experimental value. Palaniappan et al. [15] used the Box–Behnken (BB) experimental design to investigate the influences of process parameters, such as the layer thickness, printing speed, raster angle, and infill density, on the mechanical properties and failure mechanism of FDM-printed samples. Gòmez-Ortega et al. [11] employed a Taguchi orthogonal array to assess the impact of process parameters, including the extruder temperature, infill density, and wall thickness, on the tensile and flexural properties during the 3D-printing of nylon-chopped carbon fibres. Selim [16] comparatively studied the effects of production parameters on the Izod impact strength of pure PA6, PA6-15% carbon fibre, and PA6-30% glass fibre in ZX printing orientation. The Taguchi experimental design and S/N ratio were used to identify the influence of the production parameters (nozzle temperature, layer thickness, number of walls, and post-heat treatment) and the optimal experimental conditions.

With the help of experimental design methods, some studies have analysed the influence of the parameters on the mechanical properties of nylon–carbon fibre composites during FDM. However, the parameters used in the different studies varied considerably. Several studies have considered different types of factors as model inputs. For instance, some studies simultaneously considered process parameters, infilling patterns, reinforcement content, and post-processing as inputs. Consequently, the optimal results vary significantly. Therefore, establishing a quantitative model for the same class of influencing factors and properties would be helpful for industrial applications. Additionally, from the perspective of industrial applications, many engineering components can withstand bending and impact loads during operation. However, there is a lack of studies on the multi-property optimisation of the bending and impact of FDM-printed nylon–carbon-fibre composites.

In this study, a composite face-centred (CCF) experimental design and the response surface method (RSM) were employed to investigate the influence of individual and interactive parameters on the flexural and impact properties of FDM-printed short carbon fibre-reinforced nylon. Then, statistical models for the flexural strength, modulus, and impact strength were established, which were helpful for predicting and optimising the target properties of the printed sample. Finally, the comprehensive properties were optimised using the entropy weighting method and desirability function. The fractured surfaces of the bending and impact samples were characterised to investigate the fracture mechanisms. Our study provides a systematic optimisation strategy to enhance both the flexural and impact properties of FDM-printed short carbon fibre-reinforced nylon composites, making them more capable in various engineering scenarios.

## 2. Experimental Procedures

### 2.1. Material and Printing Device

A PAHT-CF filament with a diameter of 1.75 mm (purchased from Suzhou Fusi Luoke New Materials Co., Ltd., Suzhou, China) was used. PAHT is a modified nylon material with superior high-temperature performance. The weight content of chopped short carbon fibres was 15%. The length of the reinforced carbon fibre was approximately 0.6 mm, which is significantly longer than that of commonly used fibres. The melting point of the filament is 237 °C, and the supplier’s recommended printing parameters are as follows: extruder temperature of 300–320 °C, bed temperature of 70–90 °C, and printing speed of 30–120 mm∙s^−1^. Prior to printing, the filament was placed in a drying box at 70 °C for 24 h. A Blade 1 Pro printer (Dongzhiying (Shenzhen) Technology Co., Ltd., Shenzhen, China) was used to fabricate the samples (Figure 1). To improve the abrasive resistance to carbon fibres and avoid nozzle clogging, a hardened steel nozzle with a diameter of 0.6 mm was used.

The main aim of the present study was to analyse the influence of process parameters on the mechanical performance of FDM-printed PATH-CF. Therefore, the samples were printed using a concentric pattern with 100% infill density and without the boundary walls. The thickness of layer was set as 0.1–0.3 mm, and the build orientation was along the z-direction, as shown in the inset of Figure 1a.

### 2.2. Mechanical Tests and Observation of Fracture Surface

A three-point bending test can comprehensively examine the mechanical performance of the FDM-printed samples under tensile and compressive states. The bending tests were conducted at a speed of 2 mm∙min^−1^ using 100-kN universal testing machine (C51.105, SSANS, Shanghai, China) according to ASTM D790 (Standard test methods for flexural properties of unreinforced and reinforced plastics and electrical insulating material. 2007), as shown in Figure 2a. The dimensions of the sample were 127 × 12.7 × 3.2 (mm^3^). The ultimate flexural strength (*σ*) and flexural modulus (*E_f_*) can be calculated by the following equations [17]:(1)σ=3FmaxL2bfhf2(2)Ef=L3θ4bfhf3

Here, *F_max_* and *L* represent the maximum load applied and span length, respectively; *b_f_* and *h_f_* refer to the width and height of the specimen, respectively; *θ* is the slope of the elastic deformation phase in the load–displacement curve, as shown in Figure 2b.

The Charpy impact strengths of the samples were determined using a pendulum impact tester (MDS-5D-CM, MADSUR Analytical Instrument Co., Ltd., Shanghai, China) according to the ISO 179-1 standard (Plastics―Determination of Charpy impact properties, 2023), as shown in Figure 2c. The dimensions of the unnotched samples were 80 × 10 × 4 (mm^3^). The impact tests were performed using a 2-J hammer in a laboratory environment with a temperature of 25 °C and relative humidity of 40% [3,18]. The impact energy to the fracture sample *E_C_* was automatically recorded, which included the energy losses due to bearing friction and air resistance [17]. The impact toughness (*α_cU_*) of the unnotched specimen can be calculated from the following equation:(3)αcU=EChi×bi
where *h_i_* and *b_i_* are the thickness and width of the sample, respectively.

Laser scanning confocal microscopy (OLYMPUS OLS-5100 SAF, Tokyo, Japan) and scanning electron microscopy (SEM, TESCAN MIRA LMS, Brno, Czech Republic) were employed to observe the fractured surface morphology of the samples during the bending and impact tests. Prior to SEM observation, the samples were sputter-coated with gold.

### 2.3. Experiment Design

RSM is a statistical method used to obtain quantitative relationships between variables and responses. Two types of experimental designs are commonly used in conjunction with RSM: the BB and central composite design (CCD). BB design is suitable for analysing in interaction effects within the predefined factor range via relatively less experimental runs, whereas CCD is more suitable for searching the optimal solution, specifically for modelling a complicated relationship between variables and responses [19]. Podstawczyk et al. [20] compared CCF, BB, and full factorial design for modelling and optimisation by RSM. The analysis of the statistical results revealed that the best model is attained on the basis of CCD. Limited by the range of variables and to reduce the experiment runs, CCF (*α* = 1) was employed in this study. The total number of experimental runs (*N*) was calculated using Equation (4):*N* = 2^*k*^ + 2*k* + *n_c_*(4)
where *k* and *n*_c_ represent the number of variables and the centre point (i.e., the replicate experiment), respectively. Moreover, the four variables were the extruder temperature (X_1_), bed temperature (X_2_), printing speed (X_3_), and layer thickness (X_4_), and the three responses were the flexural strength (Y_1_), flexural modulus (Y_2_), and impact strength (Y_3_). The number of centre points was seven, and the total number of experimental runs was 31. Three replicates of each flexural and impact test were conducted, and the experimental results are summarised in Appendix A Table A1. The mean values were used for modelling analyse. Analysis of variance (ANOVA), RSM result presentation, and multi-response optimisation were conducted using the commercial statistical software, Design Expert 13.

## 3. Results and Discussions

### 3.1. Effect of Process Parameters on the Flexural Strength (Y_1_)

ANOVA was conducted at a 95% confidence level to evaluate the significance of the regression model, individual parameters, interactive terms, and the second order terms. A stepwise regression method was used to establish the regression models for the responses. Significant terms with *p*-value < 0.05 were retained in the model. Table 1 displays the ANOVA results for the flexural strength. The higher F-value (34.2) and lower *p*-value (<0.0001) for the model indicate that the quadratic model is suitable for predicting the flexural strength within the studied parameters, as shown in Equation (5). The coefficients of determination R^2^ and adjusted R^2^ were 0.9526 and 0.8985, respectively. The higher adequate precision of 27.6545, which should be higher than four [21], indicates that the regression model can accurately predict the flexural strength. In addition, the degree of significance of the process parameters on the flexural strength followed the order of X_3_ (printing speed) > X_2_ (bed temperature) > X_4_ (layer thickness) > X_1_ (extrusion temperature).Y_1_ = 69.50759 + 0.245994X_1_ − 3.52034X_2_ − 0.454230X_3_ + 540.00507X_4_ + 0.00508X_2_X_3_ − 6.03806X_2_X_4_ − 0.306764X_3_X_4_ + 0.029644X_2_^2^(5)

The relationship between the experimental and predicted results is illustrated in Figure 3a. Both values are close to the line of best fit, indicating the excellent predictive ability of the model. The perturbation curves for the flexural strength are shown in Figure 3b. As the X_3_ (printing speed) increases from 30 mm∙s^−1^ to 120 mm∙s^−1^, the flexural strength decreases significantly. A higher printing speed leads to a higher cooling rate, which results in more imperfections and weaker bond adhesion [22]. However, an inverse relationship exists among X_1_ (extrusion temperature), X_4_ (layer thickness), and flexural strength. Furthermore, with an increase in X_2_ (bed temperature), the flexural strength initially decreased slightly and then increased significantly. A higher bed temperature improves the adhesion between the bottom layers and printer platform, thus reducing the delamination and warpage of the part [23]. For the loading direction perpendicular to the interlayer direction (X–Y plane), a large layer thickness increased the bonding area between the filaments within a layer. This is conducive to enhancing the tensile strength [24], thereby improving the bending strength.

The contour plots of the significant interaction terms are shown in Figure 4. The other two parameters were set as constant at the middle level. Figure 4a shows that the combination of a higher bed temperature and lower printing speed achieves a higher flexural strength. A higher bed temperature and lower printing speed provide more heating energy and a longer heating period to improve interlayer and inter-bead adhesion [25]. Figure 4b shows that a higher bed temperature and thinner layer resulted in a higher flexural strength. Figure 4c shows that a slower printing speed and higher layer thickness lead to a higher flexural strength. In general, a higher layer thickness has a negative impact on the surface roughness and mechanical properties of the sample [26]. In the context of the three-point bending test, the bottom surface of the sample experienced the highest tensile stress, whereas the upper surface was subjected to compressive stress. Normally, failure is initiated in the bottom layers and propagates transversely through the sample [11]. It is assumed that the quality of the bottom layers plays a crucial role in the flexural strength, whereas some imperfections between the layers have a marginal impact.

### 3.2. Effect of Process Parameters on Flexural Modulus (Y_2_)

The ANOVA results for the flexural modulus are shown in Table 2. A higher F-value (27.11), lower *p*-value (<0.0001), and nonsignificant lack-of-fit (0.194) indicate a better fit of the regression model for the flexural modulus, as shown in Equation (6). Three significant individual parameters affect the flexural modulus. According to the F-values and *p*-values, the order of influence of the three parameters was X_4_ (layer thickness) > X_3_ (printing speed) > X_1_ (extrusion temperature). The coefficient of correlation R^2^ for the regression model was 0.9401 and the adjusted R^2^ was 0.9054, revealing that the predicted values were very close to the experimental values, as shown in Figure 5a.Y_2_= − 315,970 + 2135.84902X_1_ − 268.15335X_2_ − 95.93829X_3_ + 61,787.37981X_4_ + 0.967281X_1_X_2_ + 0.231993X_1_X_3_ − 149.26438X_1_X_4_ + 0.253671X_2_X_3_ − 254.00938X_2_X_4_ − 3.5294X_1_^2^ + 21,558.48276X_4_^2^(6)

Figure 5b shows the influence of various process parameters on the flexural modulus. In the three-point bending test, the sample exhibited complex stress. The bottom of the sample was subjected to tensile stress, whereas the top layers, where loading was applied, were under compressive stress. Generally, cracks initiate at the bottom of the sample and then propagate through the thickness of the sample. When cracks extend to the inter-layer delamination, crack propagation may deflect or stop [11]. Furthermore, the orientation of the chopped fibres is a critical factor in the strength and modulus of anisotropic composite materials [27]. Within the studied parameter ranges, as the layer thickness (X_4_) increased, the flexural modulus gradually increased until the reference point and then noticeably increased. It is assumed that a thicker layer has a higher section area of the strand (defined by the width and height of the strands) compared to a thinner layer, which is helpful in bearing a higher load [26]. In addition, proper alignment of chopped fibres in thicker layers is conducive to improving the flexural modulus [28]. The printing speed (X_3_) and flexural modulus are inversely proportional, that is, an increase in the printing speed results in a decrease in the flexural modulus. With an increase in the extrusion temperature (X_1_), the flexural modulus initially increased until the midpoint and then decreased. A higher extrusion temperature increases the fluidity of the heated filament and the reheating temperature of the previously deposited layers, consequently enhancing layer adhesion. Increasing the nozzle temperature further increases the amount of carbon fibres within the extruded thermoplastics [29]. Consequently, more defects may occur in the printed part [11], thus substantially reducing the flexural modulus.

Contour plots of the two-way interaction of the process parameters for the flexural modulus are shown in Figure 6; the other two parameters were set constant at the middle level. Owing to the various factors influencing interlayer adhesion, many complex interactions between the parameters are determined by the flexural modulus. Figure 6a shows that the middle nozzle temperature led to a higher flexural modulus regardless of the bed temperature. Figure 6b shows the extrusion temperature between 305 °C and 315 °C combined with slower printing speed result in the maximum flexural modulus. Figure 6c shows that a larger layer thickness within a wide range of extrusion temperatures can result in a higher flexural modulus. Figure 6d shows that a lower printing speed is conducive to the improvement of the flexural modulus, whereas the bed temperature has a marginal impact. The two-way interaction between bed temperature and layer thickness is shown in Figure 6e. At medium to high layer thicknesses, the bed temperature had a marginal influence on the flexural modulus. It is worth noting that a lower bed temperature and thinner layer are detrimental to the flexural modulus and should be avoided in the process parameter design.

### 3.3. Effect of Process Parameters on Impact Strength (Y_3_)

The ANOVA results for the impact strength are listed in Table 3. The *p*-values and F-values of the model were <0.0001 and 25.22, respectively. The *p*-value of the lack of fit was 0.4704. This indicates that the fitting model has a high reliability and prediction accuracy, as shown in Equation (7). It was also found that these three process parameters had a notable influence on the impact strength. According to the *p*-values and F-values, the order of the influential parameters was X_4_ (layer thickness) > X_3_ (printing speed) > X_2_ (bed temperature).Y_3_= − 72.07277 + 0.338696X_1_ + 0.045223X_2_ + 0.589702X_3_ + 170.05605X_4_ − 0.002122X_1_X_3_ − 0.827944X_1_X_4_ + 0.186599X_3_X_4_ + 127.83598X_4_^2^(7)

Figure 7a shows that the predicted and experimental values fluctuate close to the diagonal line. The determination coefficients R^2^ and adjusted R^2^ were 0.9017 and 0.8985, respectively. Additionally, the adequate precision of the model was 23.274, which should be higher than four [30], demonstrating that the fitting model can be used to predict the impact strength within the experimental range.

Figure 7b shows the variation in the impact strength with various process parameters. The impact strength decreased significantly as the layer thickness (X_4_) increased until the reference point was reached. Then it gradually decreased as the layer thickness increased from 0.2 mm to 0.3 mm. Thinner layers improve the stiffness and strength of printed parts owing to their more homogeneous heat distribution and smaller void density [12]. According to the inversely proportional relationship between the layer thickness and the number of layers, samples with thinner layers have a higher number of layers, thus leading to an improvement in the impact strength perpendicular to the building direction (z-axis) [31]. When the printing speed (X_3_) increases from 30 to 120 mm∙s^−1^, the impact strength decreases significantly. A higher printing speed leads to a partially melted filament and higher cooling rate, resulting in larger voids and weaker interlayer adhesion, thereby reducing the impact strength. With an increase in both the extrusion and bed temperatures, the impact strength increases slightly.

The significant interaction terms are displayed in the contour plots shown in Figure 8. The other two factors were set as constants at the middle level. The direction of impact loading was perpendicular to the build orientation; therefore, the status of the layer adhesion and the strength of the extruded filament simultaneously governed the impact strength of the sample. The two-way interaction between the extrusion temperature and printing speed is shown in Figure 8a. At lower extrusion temperatures, the printing speed had a marginal influence on the impact strength, which is consistent with Beylergil’s results [3]. The maximum impact strength was achieved at the highest extrusion temperature and slowest printing speed. Figure 8b shows the interaction between the extrusion temperature and layer thickness. At medium to low layer thicknesses, the extrusion temperature had a slight effect on the impact strength. Regardless of the extrusion temperature, the highest impact strength was achieved for the thinner layers. A thinner layer is helpful for reducing defects and strengthening layer adhesion [32]. Figure 8c shows that with a thinner layer and at a slower printing speed, the impact strength was higher. However, higher layer thickness and printing speed decreased the impact strength.

### 3.4. Correlation Analysis Between the Process Parameters and Mechanical Properties

Pearson correlations between the process parameters and the mechanical properties are displayed in Figure 9. There are none or extremely weak linear relationship among the four process parameters. Based on the values of correlation coefficient, the degree of linear correlation between the process parameters and mechanical properties can be determined. On the other hand, the importance of process parameters on the mechanical properties can also be ranked, showing the same trends as aforementioned based on ANOVA. In addition, the correlation coefficient between the flexural strength and modulus is high at 0.6673, indicating a strong linear relationship. Although the flexural strength and modulus describe the different abilities of component during the bending test, the positive correlation can be used to decrease the number of objectives, leading to a simplification of the optimisation process [33]. The correlation coefficient between flexural modulus and impact strength is −0.3247, indicating a negative linear relationship.

### 3.5. Multi-Response Optimisation and Validation

A desirability function approach was employed to determine the optimal parameter combination that yielded the best comprehensive mechanical properties. The desirability function maps the function of individual responses to an interval [0, 1] that represents the degree of decision makers, where 1 is highly desirable, and 0 is unacceptable. Subsequently, based on the weights of the individual responses, a comprehensive desirability function was established that integrates the aforementioned individual desirability function. Considering the optimisation problem of multiple responses, the entropy weighting method can be used to objectively assign the weight of the response, avoiding the influence of decision-maker bias [34,35,36]. The fundamental principle of the entropy weighting method is that a higher weight index is more important or provides more information than a lower weight index [37]. The detailed calculation procedures for the entropy weights are presented in Appendix B. The weights of flexural strength, flexural modulus, and impact strength were 0.2584, 0.3180, and 0.4236, respectively. The predicted results of the best comprehensive mechanical properties are 169.881 MPa for the flexural strength, 9249.11 MPa for the flexural modulus, and 29.659 kJ∙m^−2^ for the impact strength, achieved under the parameter combination of extrusion temperature of 318.12 °C, bed temperature of 89.9997 °C, printing speed of 30.01 mm∙s^−1^, and layer thickness of 0.1 mm. The overall desirability was 0.935, as shown in Figure 10. Experimental validation was conducted using slightly adjusted parameters, as listed in Table 4. The percentage errors between the predicted and actual results were 0.890% for the flexural strength, 0.996% for the flexural modulus, and 7.856% for the impact strength. This indicates that the desirability function combined with the entropy weighting method can accurately predict the comprehensive mechanical properties, and the results are more robust and highly integrated compared with previously reported results, as shown in Table 5. In addition, there are differences in the short carbon-fibre-reinforced nylon filaments from various suppliers; therefore, the reported results show significant discrepancies.

The morphologies of the fractured surfaces in the three-point bending and impact tests of the samples fabricated using the optimal combination of the process parameters are shown in Figure 11. Figure 11a shows the occurrence of obvious stress-whitening phenomena on the tensile side of the cross-section of the sample in the bending test. This may be caused by a change in the crystallisation orientation of the nylon molecular chains and the movement of the molecular chains under tensile loading [40]. Dimples caused by tear fractures were also observed on the tensile side. A few bright spots appeared on the compressed side. This is probably due to light scattering caused by crazing of the cross-section of the broken filaments. In addition, Figure 11b shows many pores and gaps between the carbon fibre and nylon matrix, and fibre pull-outs on the fractured surface. The formation of voids can be attributed to the increased viscosity of polymer composite melt, because the addition of short carbon fibres affects the rheology of the polymer [41]. In addition, the different coefficient of thermal expansion of the carbon fibre and nylon matrix results in the weak bonding interface between them [3]. Thus, some the fibre is prone to be pulled out under the tensile loading. Although the force–displacement curve measured during the three-point bending process does not have an obvious yield stage, by combining with the fracture morphology, it can be inferred that the failure mode of the specimen is the coexistence of brittle and ductile failure [42]. Figure 11c,d show large undulations on the fractured surface of the damaged sample during the impact test. Slight stress whitening, numerous bright spots, and pulled-out carbon fibres were observed. It was inferred that the nylon filaments were perpendicular to the load direction, and the interlayer bonding surfaces and chopped carbon fibres jointly bore the impact load.

## 4. Conclusions

This study focused on optimising the FDM process parameters for short carbon fibre-reinforced nylon and evaluated various mechanical properties, including flexural strength and modulus, through a three-point bending test and Charpy impact strength through a pendulum impact test. The key process parameters included the extrusion and bed temperatures, printing speed, and layer thickness. The CCF experimental design and RSM were used to construct the statistical relationship between the process parameters and properties and to investigate the influences of individual and interaction terms. The entropy weighting method and desirability function were employed to optimise multiple mechanical properties. The following conclusions were drawn:(1)The flexural strength of the FDM-printed parts varied in the range of 134.253–175.123 MPa under the studied parameters. All four parameters had a significant influence on the flexural strength, and the order of the degree of influence was X_3_ (printing speed) > X_2_ (bed temperature) > X_4_ (layer thickness) > X_1_ (extrusion temperature).(2)The flexural modulus of the FDM-printed parts varied in the range of 7836.79–10,213.2 MPa. The three process parameters had a notable impact on the flexural modulus, and the ranking of the degree of influence was as follows: X_4_ (layer thickness) > X_3_ (printing speed) > X_1_ (extrusion temperature). The bed temperature had no significant effect.(3)The impact strength fluctuated in the range of 17.372–29.701 kJ∙mm^−2^. The three process parameters had a significant influence on the impact strength, and the order of the degree of influence was X_4_ (layer thickness) > X_3_ (printing speed) > X_2_ (bed temperature). The extrusion temperature has a less significant influence.(4)The optimal comprehensive mechanical properties obtained from the desirability function outperformed those reported in the literature. The predicted outcomes had a small deviation in the range of 0.890–7.856% compared with the experimental values.(5)The failure mode of the FDM-printed short carbon fibre-reinforced nylon was a combination of brittle and ductile modes.(6)Although the filament suppliers recommend ranges of printing parameters, it still necessitates modelling the relationship between the printing parameters and the mechanical properties. The statistical model can help to set the proper printing parameters, specifically aiming to achieve the designed mechanical property in the engineering context.

## Figures and Tables

**Figure 1 polymers-17-01872-f001:**
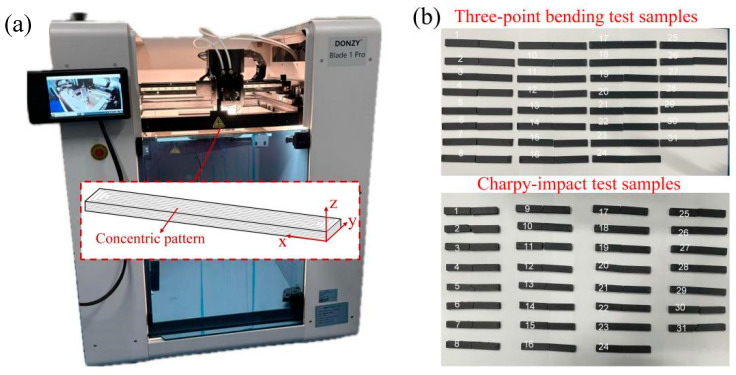
(**a**) FDM-printing device, and (**b**) pictures of test samples.

**Figure 2 polymers-17-01872-f002:**
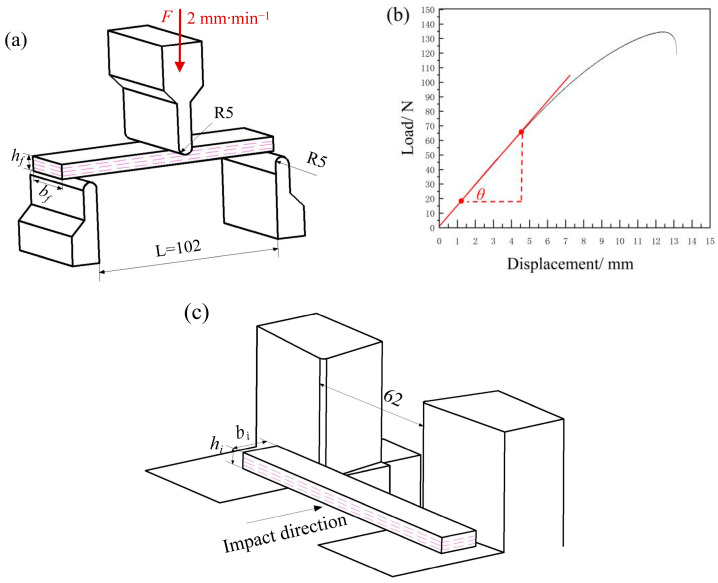
Schematic of (**a**) three-point bending test, (**b**) load–displacement curve, and (**c**) Charpy impact test.

**Figure 3 polymers-17-01872-f003:**
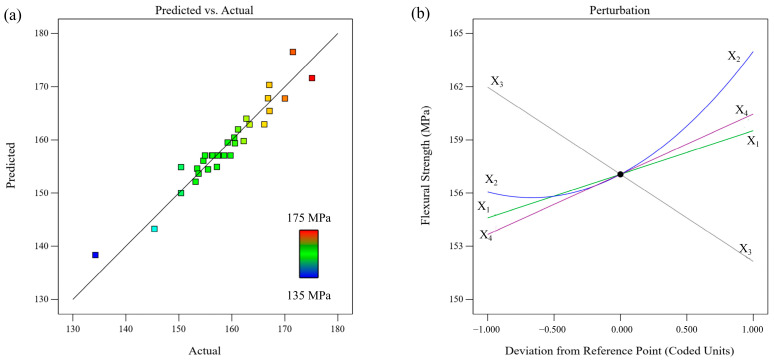
(**a**) Experimental vs. predicted values, and (**b**) perturbation plot for flexural strength.

**Figure 4 polymers-17-01872-f004:**
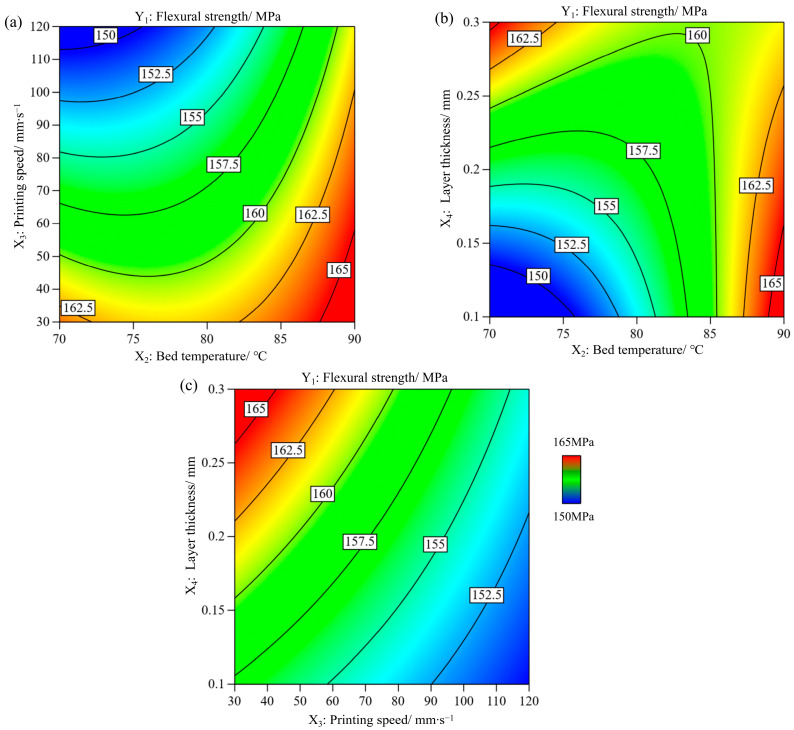
Contour plots for flexural strength variation with (**a**) bed temperature and printing speed, (**b**) bed temperature and layer thickness, and (**c**) layer thickness and printing speed.

**Figure 5 polymers-17-01872-f005:**
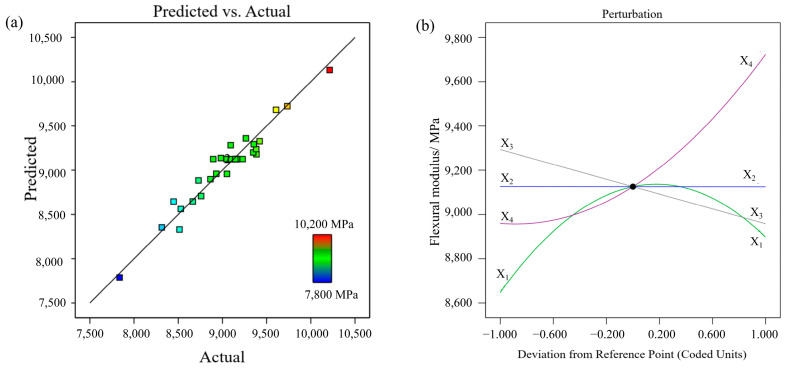
(**a**) Experimental vs. predicted values, and (**b**) perturbation plot for flexural modulus.

**Figure 6 polymers-17-01872-f006:**
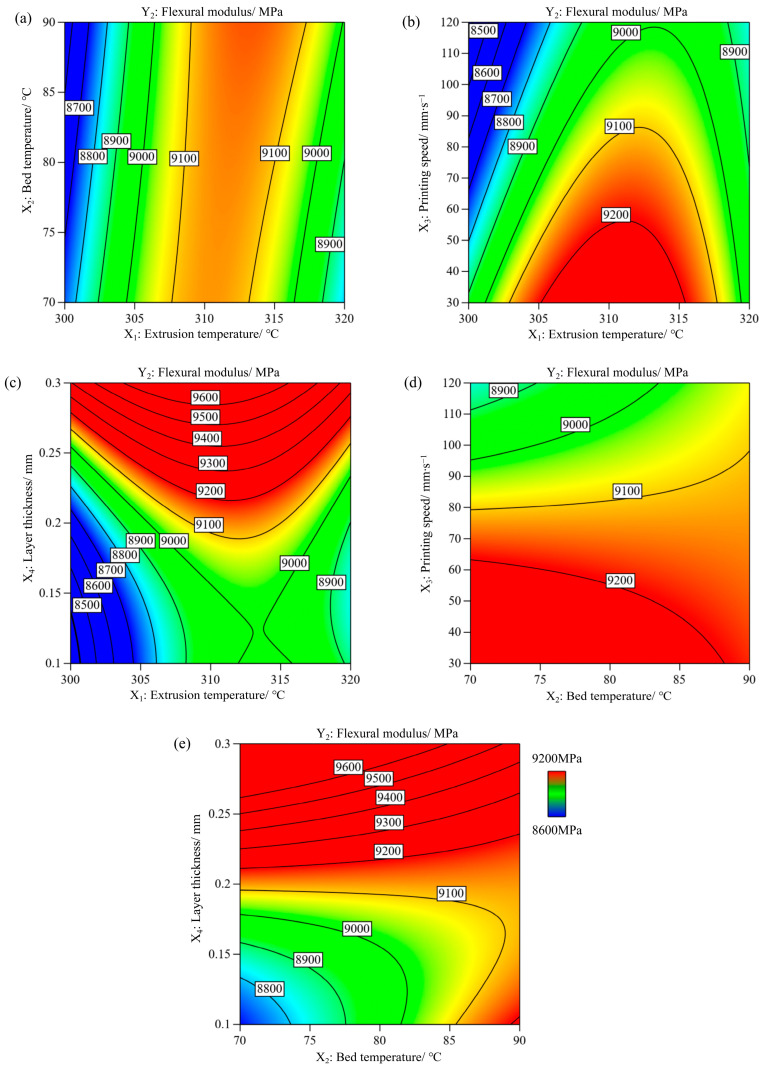
Contour plots for flexural modulus variation with (**a**) extrusion temperature and bed temperature, (**b**) extrusion temperature and printing speed, (**c**) extrusion temperature and layer thickness, (**d**) bed temperature and printing speed, and (**e**) bed temperature and layer thickness.

**Figure 7 polymers-17-01872-f007:**
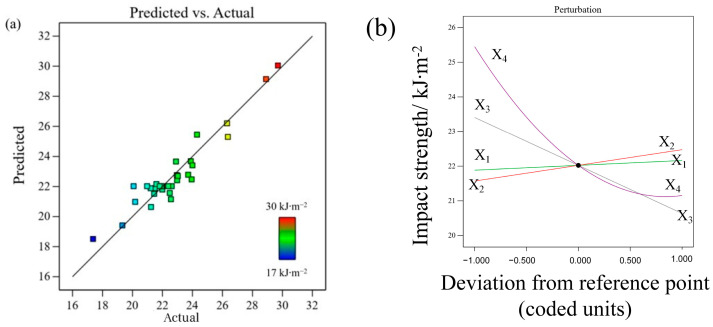
(**a**) Experimental vs. predicted values, and (**b**) the perturbation plot for impact strength.

**Figure 8 polymers-17-01872-f008:**
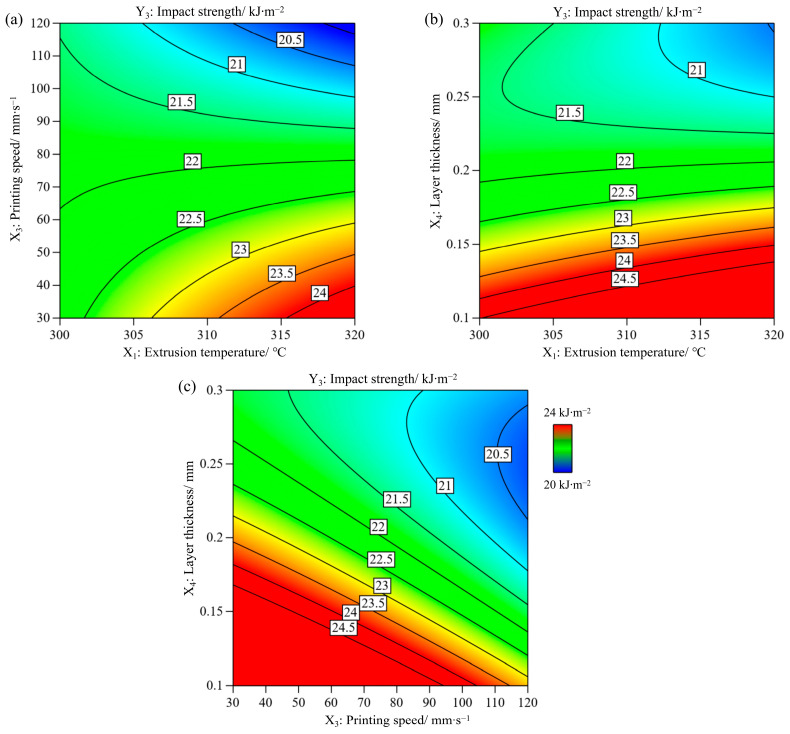
Contour plots for impact strength variation with (**a**) extrusion temperature and printing speed, (**b**) extrusion temperature and layer thickness, and (**c**) layer thickness and printing speed.

**Figure 9 polymers-17-01872-f009:**
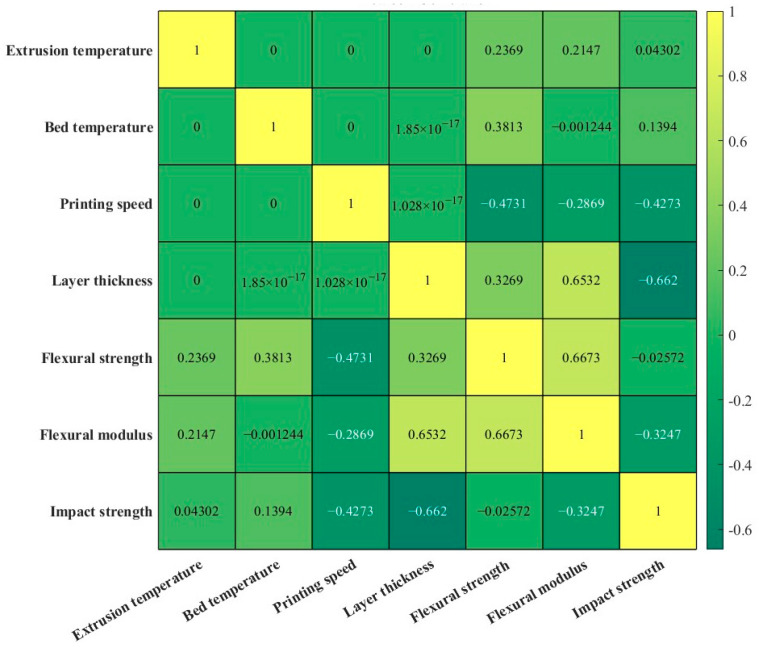
Pearson correlation coefficients of process parameters and mechanical properties.

**Figure 10 polymers-17-01872-f010:**
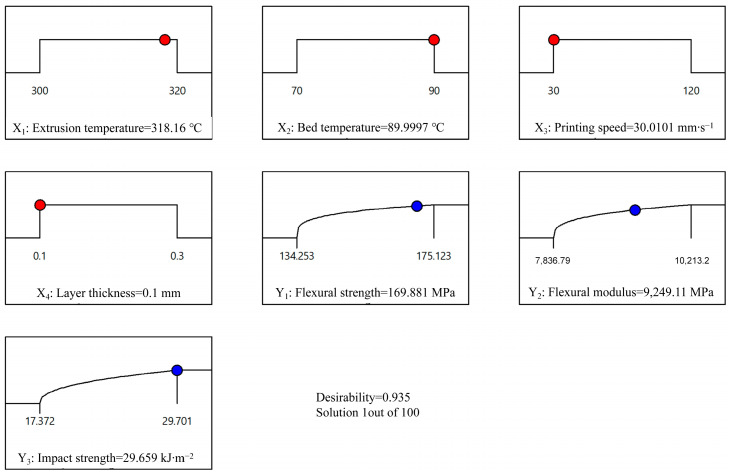
Overall desirability of the best comprehensive mechanical properties.

**Figure 11 polymers-17-01872-f011:**
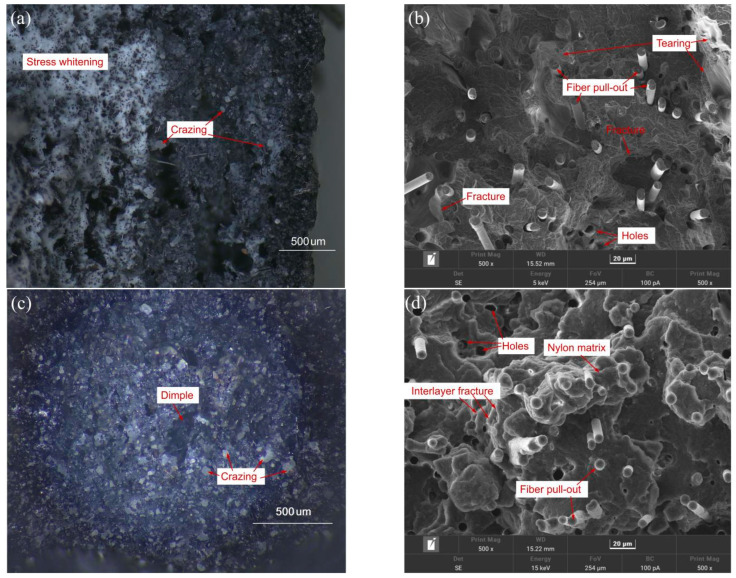
Fractured surfaces of (**a**,**b**) bending test sample, and (**c**,**d**) impact test sample.

**Table 1 polymers-17-01872-t001:** ANOVA table for flexural strength.

Source	Sum of Squares	DF	Mean Square	F-Value	*p*-Value	Contribution (%)
Model	1797.07	8	225	34.2	<0.0001	92.56
X_1_	108.92	1	109	16.59	0.0005	5.61
X_2_	282.28	1	282.28	42.98	<0.0001	14.54
X_3_	434.6	1	435	66.18	<0.0001	22.38
X_4_	207.5	1	208	31.6	<0.0001	10.69
X_2_X_3_	83.61	1	83.6	12.73	0.0017	4.31
X_2_X_4_	583.33	1	583	88.82	<0.0001	30.04
X_3_X_4_	30.49	1	30.5	4.64	0.0424	1.57
X_2_^2^	66.33	1	66.3	10.1	0.0044	3.42
Residual	144.48	22	6.57	-	-	7.44
Lack of Fit	129.31	16	8.08	3.2	0.0791	6.66
Pure Error	15.17	6	2.53	-	-	0.78
Cor Total	1941.55	30	-	-	-	100

R^2^: 0.9526, adjusted R^2^: 0.8985, adequate precision: 27.6545.

**Table 2 polymers-17-01872-t002:** ANOVA table for flexural modulus.

Source	Sum of Squares	DF	Mean Square	F-Value	*p*-Value	Contribution (%)
Model	5.78 × 10^6^	11	5.26 × 10^5^	27.11	<0.0001	93.98
X_1_	2.84 × 10^5^	1	2.84 × 10^5^	14.62	0.0011	4.62
X_2_	9.52	1	9.52	0.0005	0.9826	0.00
X_3_	5.06 × 10^5^	1	5.06 × 10^5^	26.11	<0.0001	8.23
X_4_	2.62 × 10^6^	1	2.62 × 10^6^	135.36	<0.0001	42.60
X_1_X_2_	1.50 × 10^5^	1	1.50 × 10^5^	7.72	0.012	2.44
X_1_X_3_	1.74 × 10^5^	1	1.74 × 10^5^	9	0.0074	2.83
X_1_X_4_	3.57 × 10^5^	1	3.57 × 10^5^	18.39	0.0004	5.80
X_2_X_3_	2.09 × 10^5^	1	2.09 × 10^5^	10.75	0.0039	3.40
X_2_X_4_	1.03 × 10^6^	1	1.03 × 10^6^	53.25	<0.0001	16.75
X_1_^2^	4.32 × 10^5^	1	4.32 × 10^5^	22.3	0.0001	7.02
X_4_^2^	1.61 × 10^5^	1	1.61 × 10^5^	8.32	0.0095	2.62
Residual	3.68 × 10^5^	19	19,385.45	-	-	5.98
Lack of Fit	3.01 × 10^5^	13	23,122.83	2.05	0.194	4.89
Pure Error	67,726.78	6	11,287.8	-	-	1.10
Cor Total	6.15 × 10^6^	30	-	-	-	100

R^2^: 0.9401, adjusted R^2^: 0.9054, adequate precision: 27.0435.

**Table 3 polymers-17-01872-t003:** ANOVA table for impact strength.

Source	Sum of Squares	DF	Mean Square	F-Value	*p*-Value	Contribution (%)
Model	171	8	21.4	25.22	<0.0001	90.27
X_1_	0.351	1	0.351	0.4142	0.5265	0.19
X_2_	3.68	1	3.68	4.35	0.0488	1.94
X_3_	34.6	1	34.6	40.86	<0.0001	18.26
X_4_	83	1	83	98.08	<0.0001	43.81
X_1_X_3_	14.6	1	14.6	17.24	0.0004	7.71
X_1_X_4_	11	1	11	12.96	0.0016	5.81
X_3_X_4_	11.3	1	11.3	13.33	0.0014	5.96
X_4_^2^	12.3	1	12.3	14.57	0.0009	6.49
Residual	18.6	22	0.847	-	-	9.82
Lack of Fit	14	16	0.875	1.13	0.4704	7.39
Pure Error	4.63	6	0.772	-	-	2.44
Cor Total	189.44	30	-	-	-	100

R^2^: 0.9017, adjusted R^2^: 0.8985, adequate precision: 23.274.

**Table 4 polymers-17-01872-t004:** Validation experiment of the predicted results.

Optimal Process Parameter Combination				Mechanical Properties			
Extrusion Temperature/°C	Bed Temperature/°C	Printing Speed/ mm∙s^−1^	layer Thickness/mm		Flexural Strength/mpa	Flexural Modulus/ mpa	Impact Strength/ kJ∙m^−2^
318	90	30	0.1	1	173.107	9484.739	27.785
				2	166.736	9321.681	26.561
				3	165.264	9217.148	27.642
				Average value	168.369	9341.189	27.329

**Table 5 polymers-17-01872-t005:** Comparison of flexural and impact properties of FDM-printed short carbon fibre reinforced nylon between the literature and this work.

Mechanical Properties	Process Parameters		References
	Variable Factors	Fixed Factors	
Maximum flexural strength: 67.36 ± 6.84 MPa Maximum flexural modulus: 2783.35 ± 346.9 MPa	Extrusion temperature: 230–280 °C, wall thick: 0.8–1.6 mm, infill density: 33–99%	Nozzle diameter: 0.4 mm, layer thickness: 0.2 mm, printing speed: 50∙mm∙s^−1^, infill pattern: rectilinear	[11]
Maximum flexural strength: 119.9 MPa Maximum flexural modulus: 3038 MPa	Printing speed: 30–50 mm∙s^−1^, layer thickness: 0.1–0.2 mm, extrusion temperature: 270–290 °C, build orientation: flat, on-edge	Infill pattern: rectilinear, infill density: 100%, raster angle: ±45°, bed temperature: 80 °C, nozzle diameter: 0.4 mm	[38]
Maximum flexural strength: 25.89 MPa Maximum impact strength: 0.72 kJ∙mm^−2^	Layer thickness: 0.1–0.3 mm, printing speed: 40–60 mm∙s^−1^, infill density: 60–100%	Not mentioned	[15]
Maximum impact strength: 10.54 kJ∙mm^−2^	Extrusion temperature: 240–260 °C, printing speed: 20–40 mm∙s^−1^, infill density: 80–100%, raster angle: 0–75°	Bed temperature: 80 °C, Nozzle diameter: 0.4 mm, layer thickness: 0.2 mm	[3]
Maximum flexural strength: 87 MPa Maximum impact strength: 12.5 kJ∙mm^−2^	Layer thickness: 0.07–0.2 mm, infill density: 50–100%, raster angle: 0–90°	Extrusion temperature: 215 °C, bed temperature: 110 °C, infill pattern: zig-zag	[39]
Maximum impact strength: 6.258 kJ∙mm^−2^	Extrusion temperature: 245–275 °C, layer thickness: 0.15–0.45 mm, wall line count: 0–2, heat treatment time: 0–160 min	Nozzle diameter: 0.6 mm, initial layer height: 0.25 mm, infill density: 100%, infill pattern: lines, bed temperature: 70 °C, printing speed: 40 mm∙s^−1^	[16]
Maximum flexural strength = 175.123 MPa, Maximum flexural modulus = 10,213.2 MPa, Maximum impact strength = 29.701 kJ∙mm^−2^	Extrusion temperature: 300–320 °C, bed temperature: 70–90 °C, printing speed: 30–120 mm∙s^−1^, layer thickness: 0.1–0.3 mm,	Infill pattern: concentric, infill density: 100%, nozzle diameter: 0.6 mm	This work
Optimal comprehensive mechanical properties: flexural strength = 168.369 MPa, flexural modulus = 9341.189 MPa, impact strength = 27.329 kJ∙mm^−2^			

## Data Availability

All data supporting the findings of this study are included in the article.

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
