# Peer review of "Modelling and Optimisation of FDM-Printed Short Carbon Fibre-Reinforced Nylon Using CCF and RSM"

_polymers, 2025, doi:10.3390/polym17131872_

Round 1

Reviewer 1 Report

Comments and Suggestions for Authors

In this manuscript, Modelling and Optimization of FDM printed short carbon fibre-reinforced nylon using CCF and RSM, the authors employed a central composite face-centred (CCF) design with four factors, three levels, and the response surface method to reinforce Nylon using short carbon fibres. The manuscript can be accepted after minor changes suggested in the following comments:

  1. More details about the CCF and RSM should be added to facilitate direct comparison with Box-Behnken or full factorial designs for RSM, even though the number of runs is less in Box-Behnken.
  2. Could you consider adding more levels for any of the parameters studied, extruder temperature (X1), bed temperature (X2), printing speed (X3), and layer thickness (X4), and the three responses were the flexural strength (Y1), flexural modulus (Y2), and impact strength (Y3), which could improve the accuracy of the model?
  3. A three-point bending test can comprehensively examine the mechanical performance of the FDM-printed samples under tensile and compressive states. I suggest adding at least 7 points for testing and improving center-to-edge uniformity and confidence.
  4. Are there any tradeoffs observed in mechanical properties, especially at different speeds? How will different speeds impact the mechanical properties?
  5. How would you explain or justify the strong influence of printing speed on both flexural strength and impact strength?
  6. The adjusted R2 is slightly lower than R2, which is acceptable, but would overfitting be adjusted?
  1. Figures 4 and 6, contour plots show local optima. Did you verify these points experimentally, which could significantly improve confidence in the model prediction?
  1. It would also be a good idea to provide an EDS analysis to find out the uniformity of the surface, which could significantly impact the mechanical properties.

Author Response

Response 1: BB design is suitable for analyzing in interaction effects within the predefined factor range via relatively less experimental runs, whereas CCD is more suitable for searching the optimal solution, specifically for modeling complicated relation between variables and responses[19]. Podstawczyk et al.[20] compared CCD, BB, and full factorial design for modeling and optimization by RSM. The analysis of the statistical results revealed that the best model is that attained on the basis of CCD.

Comment 2: Could you consider adding more levels for any of the parameters studied, extruder temperature (X1), bed temperature (X2), printing speed (X3), and layer thickness (X4), and the three responses were the flexural strength (Y1), flexural modulus (Y2), and impact strength (Y3), which could improve the accuracy of the model?

Response 2: In theory, more levels of factors can yield more experimental data, thus increasing the prediction accuracy. DOE has proved to be cost-effective by lowering the number of experimental runs to explore the optimum parameters. For a CCD with four factors, three levels is enough to fit a second order model and achieve acceptable prediction accuracy.

Comment 3: A three-point bending test can comprehensively examine the mechanical performance of the FDM-printed samples under tensile and compressive states. I suggest adding at least 7 points for testing and improving center-to-edge uniformity and confidence.

Response 3: In this study, we conducted the flexural test according to ASTM D790, which is using three-point bending test. In the future study, we will consider the reviewer’s suggestion, using more points for testing.

Comment 4: Are there any tradeoffs observed in mechanical properties, especially at different speeds? How will different speeds impact the mechanical properties?

Response 4: From our present experimental results, the printing speed has negative effect on both flexural and impact properties. There are numerous parameters in FDM process, in my opinion, if involving more printing parameters in a study, when doing multiple objective optimization, it probably needs to make trade-offs depending on the optimization goals.

Comment 5: How would you explain or justify the strong influence of printing speed on both flexural strength and impact strength?

Response 5: According to ANOVA results, the influence significance of the factors can be ranked on the basis of F-value and P-value. In addition, the contribution of the factors can also be referred, as shown in last column in Table 1, 2 and 3.

Comment 6: The adjusted R2 is slightly lower than R2, which is acceptable, but would overfitting be adjusted?

Response 6: When assessing the goodness of fit of one model, R² is more intuitive. In addition, when the sample size is less than 50, the acceptable difference between R² and Adjusted R² is approximately 0.1.

Comment 7: Figures 4 and 6, contour plots show local optima. Did you verify these points experimentally, which could significantly improve confidence in the model prediction?

Response 7: In the present study, we intend to make multi-response optimization. Therefore, we did not optimize the individual mechanical properties, thus the local optima is not verified by experiment. In the future study, we will consider the reviewer’s suggestion.

Comment 8: It would also be a good idea to provide an EDS analysis to find out the uniformity of the surface, which could significantly impact the mechanical properties.

Response 8: Thanks for the reviewer’s suggestion. The elemental composition of nylon is C, N, and O, and that of carbon fibre is C, which are all classified as a light element, posing challenges for quantitative analysis by EDS. In the future study, we will consider other techniques for material testing and characterization.

Reviewer 2 Report

Comments and Suggestions for Authors

Abstract:

This article presents a systematic study of optimizing the mechanical performance of short carbon fiber-reinforced nylon by fused deposition modeling (FDM). The authors use a Central Composite Face-centered (CCF) with response surface methodology (RSM) to model the effects of four key factors (extrusion temperature, bed temperature, printing speed, and layer thickness) on flexural strength, flexural modulus, and impact strength. The authors use the desirability function and entropy weighting method to perform multi-objective optimization. The optimized mechanical properties are validated experimentally, with good agreement between experimental and predicted values. Page 3, Line 7, there is a spelling error in the word 'comparison'; please correct the spelling.

  1. In this study, a robust experimental design using Central Composite Face-centered (CCF) methodology with Response Surface Methodology (RSM) will provide the most complete examination of the main and interaction effects of the process parameters on the mechanical performance of FDM-printed short carbon fibre-reinforced nylon. This statistical method offers a fine approach to optimize multi-parameter systems and would have been performed rampantly in this study.
  2. The optimal results from the models were tested experimentally, and the low differences in predicted versus actual values (under 8%) validate the model's capacity for reliability. This adds a significant amount of credibility to the optimization framework proposed, and it validates the likelihood of some applicable use in industry.
  3. The manuscript gives a detailed description of how each process parameter (extrusion temperature, bed temperature, printing speed, and layer thickness) affects the mechanical results. The perturbation plots, contour plots, and ANOVA tables help to clarify the results and provide very accessible information for practitioners who would like to optimize FDM settings for high-strength composite parts.
  4. It is recommended to use the following paper published in Polymer Journal. Various FDM mechanisms used in the fabrication of continuous-fiber reinforced composites: a review.
  5. Although the study is well-conducted, the novelty of the work could be more clearly articulated, particularly in comparison to existing studies that have explored mechanical optimization of nylon-carbon fiber composites using similar experimental techniques. The authors should more explicitly state how their methodology or findings represent a significant advancement over previous research.
  6. The manuscript has several typographical and formatting issues, including improper cross-referencing (e.g., "Error! Reference source not found."), Inconsistent figure captions and occasionally unclear phrasing. These errors can impair the readability of the manuscript and should be remedied before publication.

Author Response

Comments 1-3 are the reviwer's comment to this study, it doesn't not to reply. The other three responses to comments 4-6 are listed as followings:

Comment 4: It is recommended to use the following paper published in Polymer Journal. Various FDM mechanisms used in the fabrication of continuous-fiber reinforced composites: a review.

Response 4: The recommended the reference is a very informative review paper. We have cited in the introduction section, reference[10].

Comment 5: Although the study is well-conducted, the novelty of the work could be more clearly articulated, particularly in comparison to existing studies that have explored mechanical optimization of nylon-carbon fiber composites using similar experimental techniques. The authors should more explicitly state how their methodology or findings represent a significant advancement over previous research.

Response 5: Thanks for reviewer’s constructive suggestion. We add a short statement at the end of introduction section, also as follows:

Our study provides a systematic optimization strategy to enhance both the flexural and impact properties of FDM-printed short carbon fibre-reinforced nylon composites, making them more capable in various engineering scenarios.

Comment 6: The manuscript has several typographical and formatting issues, including improper cross-referencing (e.g., "Error! Reference source not found."), Inconsistent figure captions and occasionally unclear phrasing. These errors can impair the readability of the manuscript and should be remedied before publication.

Response 6: Sorry for our carelessness. The Figure 11 has been renumbered in the corresponding text.

There is “Table 5 Comparison of flexural and impact properties of FDM printed short carbon fibre reinforced nylon between literature and this work” at the end of the whole manuscript, before reference list, reference[38]and [39] are only appeared in Table 5, not in the text.

Reviewer 3 Report

Comments and Suggestions for Authors

The article investigates the flexural and impact properties of FDM-printed short carbon fiber-reinforced nylon using a central composite face-centered (CCF) design and response surface methodology (RSM). The study aims to optimize four key process parameters - extrusion temperature, bed temperature, printing speed, and layer thickness - to enhance mechanical properties.
Before I can reconsider my recommendation for publication, the authors should address the following issues:
1) The experimental design section does not provide sufficient detail on the selection of parameter ranges. It would be helpful if the authors could justify their choice based on material properties or equipment constraints.
2) The manuscript contains several instances of missing or incorrect reference citations, which undermines the credibility of the discussion (e.g., "Error! Reference source not found" in Sections 3.1–3.3).
3) The discussion of results primarily focuses on describing trends in the data, but it lacks a critical evaluation of how these findings compare to or extend existing knowledge.
4) The discussion of the results focuses on describing the trends in the effects of printing speed or layer thickness, but it also lacks a critical evaluation of how these findings compare to or extend existing knowledge. Can the positive correlation between flexural strength and modulus, or the negative correlation with impact strength, guide material design?

Author Response

Comment 1: The experimental design section does not provide sufficient detail on the selection of parameter ranges. It would be helpful if the authors could justify their choice based on material properties or equipment constraints.

Response 1: Thanks for reviewer’s constructive suggestion. We explained the influence of four selected FDM process parameters on the printing quality in the introduction section, Page 2 Line73-82.

the printing temperature is a critical parameter for printed thermoplastic composite filaments because it has a significant influence on the bonding strength of the reinforcement/polymeric matrix and the interlayer adhesion. The nozzle temperature determines the molten state and fluidity of the filament, whereas the printing speed governs the cooling process. Consequently, both parameters are directly related to the heating history during the entire modelling process. The layer thickness also governs layer adhesion and surface quality (staircase effect), thus determining the strength and stiffness of the printed parts. The platform temperature determines the initial layers that stick to the platform, thus significantly influencing the warpage and shrinkage of the entire printed part[13].
Comment 2: The manuscript contains several instances of missing or incorrect reference citations, which undermines the credibility of the discussion (e.g., "Error! Reference source not found" in Sections 3.1–3.3).

Response 2: Sorry for our carelessness. We have reordered the reference number, which is at the end of the main text, before reference. In addition, reference[38]and [39] are only appeared in Table 5, not in the text.
Comment 3: The discussion of results primarily focuses on describing trends in the data, but it lacks a critical evaluation of how these findings compare to or extend existing knowledge.

Response 3: We made a comparison between the present study and reported results, and summarized them in Table 5.
Comment 4: The discussion of the results focuses on describing the trends in the effects of printing speed or layer thickness, but it also lacks a critical evaluation of how these findings compare to or extend existing knowledge. Can the positive correlation between flexural strength and modulus, or the negative correlation with impact strength, guide material design?

Response 4: We added a new section 3.4 in the text, which discussed the Correlation analysis between the process parameters and mechanical properties using Pearson correlations analysis.

Reviewer 4 Report

Comments and Suggestions for Authors

This study optimizes FDM process parameters (extrusion temperature, bed temperature, printing speed, layer thickness) for short carbon fiber-reinforced nylon (PAHT-CF) using a Central Composite Face-centered (CCF) design and Response Surface Methodology (RSM). The aim is to model and maximize flexural strength, flexural modulus, and impact strength. Key strengths include rigorous experimental design (31 runs), robust ANOVA analysis, and a novel multi-objective optimization approach combining entropy weighting with desirability functions. The optimized parameters (318°C extrusion, 90°C bed, 30 mm·s⁻¹ speed, 0.1 mm layer thickness) achieved high mechanical properties with low prediction errors (0.89–7.856%). However, several concerns regarding the work are noted:

  1. Figure 11. The text lacks a reference to this figure. Judging by the context, the authors likely made an error, and the text should reference Figure 11 instead of Figure 10.
  2. Figure 10 – The image is too small; specifically, the size of the labels/numbers needs to be increased.
  3. The authors state: "A thinner layer is helpful for reducing defects and strengthening layer adhesion." While the authors include microscopic images, they fail to discuss changes in defects within the context of these obtained images. This omission represents a missed opportunity to significantly enhance the interpretability of their work, given their access to microscopy for visualizing results.
  4. The Materials and Methods section lacks detailed information concerning microscopy. Specifically omitted are: sample preparation procedures and scanning/fluorescence microscopy parameters.
  5. A citation error ("Error! Reference source not found.") appears throughout the text.

Author Response

Comment 1: Figure 11. The text lacks a reference to this figure. Judging by the context, the authors likely made an error, and the text should reference Figure 11 instead of Figure 10.

Response 1: Sorry for our carelessness. The number of figures has been renumbered in the corresponding text.

Comment 2: Figure 10 – The image is too small; specifically, the size of the labels/numbers needs to be increased.

Response 2: Figure 10 has been redrawn.

Comment 3: The authors state: "A thinner layer is helpful for reducing defects and strengthening layer adhesion." While the authors include microscopic images, they fail to discuss changes in defects within the context of these obtained images. This omission represents a missed opportunity to significantly enhance the interpretability of their work, given their access to microscopy for visualizing results.

Response 3: Thanks for reviewer’s constructive suggestion. We add the explanation for the formation of voids in the text, and also as follows:

The formation of voids can be attributed to the increased viscosity of polymer composite melt, because the addition of short carbon fibres affects the rheology of the polymer[42]. In addition, the different coefficient of thermal expansion of the carbon fibre and nylon matrix results in the weak bonding interface between them[3]. Thus, some the fibre is prone to be pulled out under the tensile loading. 

Comment 4: The Materials and Methods section lacks detailed information concerning microscopy. Specifically omitted are: sample preparation procedures and scanning/fluorescence microscopy parameters.

Response 4: We have added the description of the preparation procedure of SEM observation in the revised manuscript. “Prior to SEM observation, the samples were sputter-coated with gold.”

Comment 5: A citation error ("Error! Reference source not found.") appears throughout the text.

Response 5: There is “Table 5 Comparison of flexural and impact properties of FDM printed short carbon fibre reinforced nylon between literature and this work” at the end of the whole manuscript, before reference list, reference[38]and [39] are only appeared in Table 5, which is at the end the main body, before reference, but are not referred not in the text.

Round 2

Reviewer 2 Report

Comments and Suggestions for Authors

The manuscript is well-revised.

Reviewer 3 Report

Comments and Suggestions for Authors

The authors have made some changes to the manuscript, and I have accepted them.